# Health Education about Lifestyle-Related Risk Factors in Gynecological and Obstetric Care: A Qualitative Study of Healthcare Providers’ Views in Germany

**DOI:** 10.3390/ijerph191811674

**Published:** 2022-09-16

**Authors:** Manuela Bombana, Michel Wensing, Lisa Wittenborn, Charlotte Ullrich

**Affiliations:** 1Department of General Medicine and Health Service Research, University Hospital Heidelberg, Im Neuenheimer Feld 130.3, 69120 Heidelberg, Germany; 2Department of Prevention, AOK Baden-Württemberg, Presselstrasse 19, 70191 Stuttgart, Germany

**Keywords:** educational intervention, health-related behaviors, gynecological and obstetric care, pregnancy, lifestyle-related risk factors

## Abstract

Objective: Lifestyle-related risk factors (LRRFs) during pregnancy and lactation are associated with a range of health problems. However, previous studies have shown a large knowledge gap among pregnant women regarding the effects of LRRFs. This study aimed to investigate the role of health education about LRRFs during pregnancy and lactation in gynecological and obstetric care from healthcare providers’ (HCPs) point of view. Methods: To explore the views of healthcare providers, a qualitative study was performed. In 2019, 22 in-depth interviews were conducted with a purposive sample of 9 gynecologists and 13 midwives. Participants came from different inpatient and outpatient care settings and from rural, urban, and socially deprived areas in southern Germany. All the interviews were tape-recorded and transcribed verbatim. A combined inductive and deductive approach was applied for data analysis. Results: Interviews with HCPs showed that they were aware of the possible impacts of LRRFs during pregnancy and lactation. They noted the importance of action, specifically among women with low socioeconomic status (SES), migrants, and women with a concerning medical history or other specific needs. However, the interviews showed that, at present, there is no standardized practice of educating patients on LRRFs in routine care. This was attributed to a lack of guidelines and time, unfavorable regulations, and undefined responsibilities. The priority of health education is lower in inpatient healthcare settings as compared to outpatient healthcare settings. HCPs apply a demand-driven healthcare approach, focusing on a woman’s medical history, needs, and personal circumstances. HCPs voiced the importance of implementing pre-conception education across different healthcare settings, garnering support from other health organizations, and setting out clearly defined responsibilities among HCPs. Conclusions: This qualitative study explored HCPs’ perspectives on health education about LRRFs during pregnancy and lactation. The results from this study emphasize the need for a central strategy for health education about LRRFs during pregnancy and lactation in gynecological and obstetric care.

## 1. Introduction

Lifestyle-related risk factors (LRRFs) during pregnancy and lactation are associated with morbidity and mortality [1,2,3,4,5,6,7,8,9,10,11,12]. Several systematic reviews have shown the effects of maternal alcohol consumption [3,4,5], tobacco smoking [5,6,7,8], coffee consumption [13,14], malnutrition [2], physical inactivity [9,10,11], oral health issues [15], and medication during pregnancy [16,17].

Moreover, several studies have found that LRRFs during lactation negatively affect offspring health outcomes [18,19]. For example, after alcohol consumption, breast milk reaches the same alcohol levels as maternal blood levels within 30 to 60 min [20]. LRRFs during pregnancy can lead to substantial individual, social, and economic burdens. For example, a child with a fetal alcohol spectrum disorder (FASD) may face diverse developmental challenges, associated with economic costs of about USD 1,926,000 across their lifespan [21]. At the same time, LRRFs are avoidable and preventable. Educating HCPs and women on LRRFs during pregnancy and lactation in the context of gynecological and obstetric care might be an effective measure to reduce the adverse effects of LRRFs during pregnancy and lactation. Studies have found that health education interventions on oral health [22,23,24], smoking [25], alcohol consumption [26,27], and physical activity [28,29] during pregnancy were effective at initiating short- and long-term health behavior changes [25,30]. The effectiveness of health education might be related to its acceptance among patients and HCPs. A study from the Netherlands showed that health education is highly appreciated by pregnant women [31]. Specifically, during pregnancy, women experience high levels of uncertainty, and thus might be comparably open to and accepting of health education; this in turn might increase efficacy. However, given the effects of LRRFs from conception onward, it might be better to attempt to reach women prior to conception.

In Germany, the main delivery routes for health education about LRRFs during pregnancy and lactation are through inpatient and outpatient gynecological and obstetric care settings. Before and during pregnancy, the majority of women in Germany consult their gynecologists at outpatient gynecological care centers [32]. Childbirth usually takes place in hospitals. Postnatal care after delivery is conducted by the gynecologist and follow-up care by the midwife until the introduction of solid food (about 5–6 months after delivery) [33].

Studies have found health education interventions on smoking cessation [34], breastfeeding knowledge [35], self-efficacy, attitudes towards breastfeeding, and perceived breastfeeding barriers [36] during lactation to be effective. In our previous study, we found large knowledge gaps among pregnant women on LRRFs during pregnancy, and thus identified an urgent need for preventive action [37]. While there is a broad consensus on the importance of health education about LRRFs during pregnancy and lactation, the practice of health education in gynecological and obstetric care remains underinvestigated.

This paper reports on a qualitative study that aimed to explore gynecological and obstetric healthcare providers’ (HCPs) attitudes and perspectives on the current practice, processes, and conditions of health education of LRRFs in gynecological and obstetric care in Germany.

## 2. Methods

### 2.1. Study Design

A qualitative study was conducted to explore and access HCPs perspective on LRRFs in gynecological and obstetric care in Germany. Data were collected through semistructured guide-based interviews with gynecologists and midwives in inpatient and outpatient gynecological and obstetric care settings in Baden-Wuerttemberg, Germany.

The topics of the interview guide were based on a literature review and then adapted after input of an expert focus group with 5 HCPs (2 gynecologist and 3 midwifes). The interview guide was then piloted for understanding, and revisions were made when appropriate. The interview guide contained the following four thematic domains concerning health education about LRRFs in gynecological and obstetric care: (1) content of health education, (2) process of health education (e.g., time taken), (3) perceived relevance of health education about LRRFs (e.g., sociodemographic or cultural differences across women), and (4) availability of guidelines. In addition, we collected sociodemographic data including age, gender, category of HCP, category of healthcare institution, area of healthcare institution, and years of experience as HCP.

The study was conducted in accordance with the Declaration of Helsinki. The study received ethical approval from the Ethics Committee of the Medical Faculty of Ruprecht Karls University Heidelberg on 20 May 2019 (S-340/2019).

### 2.2. Study Population, Recruitment and Data Collection

Sampling was conducted to reflect broad structural variance trough diversity in professions (gynecologists and midwives), healthcare settings (inpatient or outpatient gynecological or obstetric care settings), and areas (rural, urban or socially deprived). Inpatient gynecological and obstetric care settings were classified according to their care level, including maximum care providers, specialized care providers, routine care providers, basic medical care providers, and birth centers. Outpatient gynecological and obstetric care settings included practices in rural, urban, and socially deprived areas.

Lists of possible participants were compiled using public registers of the Association of Statutory Health Insurance Physicians Baden-Wuerttemberg and the Midwives Association Baden-Wuerttemberg, as well as the National Association of Statutory Health Insurance Funds. A random selection of participating gynecological and obstetric care settings was generated by a computer.

Health insurer AOK Baden-Wuerttemberg (BW) contacted 264 gynecological and obstetric care providers via mail or e-mail. Following recruitment, reminders were sent to e-mail contacts after 1 week of non-response and again after 9 days of non-response. Reminder calls were made 14 days after initial contact for all contacted HCPs with no response. A total of 21 of the contacted institutions of gynecological and obstetric care agreed to participate in the study (response rate: 8.0%) with totally 22 HCPs.

Two female interviewers with experience in qualitative research methods conducted the interviews in German between July and November 2019. Interviews were conducted at the HCPs’ workplaces (18) or by telephone (4) if preferred by the HCP. The duration of the interviews varied between 7 and 44 min. Interviews were digitally recorded and transcribed verbatim. Data collection ended when theoretical saturation was reached.

In total, 9 gynecological and 13 obstetric HCPs participated in the study. All participants gave their written informed consent for participation and audio-taping of the interviews. The participants’ anonymity and confidentiality were ensured throughout the entire study. Withdrawal from the study was possible at any time until the data analysis phase, and without requiring a reason. The participants did not receive any reimbursement for their participation in the study.

Of the 22 respondents with whom we conducted in-depth interviews, 20 were female (90.9%) and 2 were male (9.1%), with an average 20 years of experience as a gynecological or obstetric HCP (range: 2–38.5 years). Participants were, on average, 40.6 years old (range: 26–63 years). Figure 1 provides further information on the participants’ characteristics.

### 2.3. Analysis

A combined inductive and deductive approach following the basic ideas of grounded theory was applied for data analysis. Two researchers performed data analysis. In a first step, open coding was used without preset codes for data familiarization. During successive coding steps, codes were continuously developed and modified [38]. Second, axial coding was applied to refine and differentiate concepts. As per the definition, 1 category was centered and surrounded by a network of relationships. We repeatedly checked the established relationships against new data. To explore relationships between the categories, Strauss and Corbin [39] recommend examining data and codes based on a coding paradigm focusing on and relating causal conditions, context, intervening conditions, action strategies, and consequences. On the basis of this idea, an adapted coding paradigm was developed to map central findings [40]. Finally, selective coding was used to combine these coding paradigms and establish one consolidated core category on the practice of health education about LRRFs during pregnancy and lactation. Memos were written throughout the data analysis process.

Data, derived themes and subthemes, and the analytical process were discussed regularly within the research team to ensure intercoder congruity and to achieve the widest consensus possible.

The software package MAXQDA, Analytics Pro 2018 (Release 18.2.0, VERBI GmbH, Berlin, Germany) was used for the transcription of interviews, data management, and to facilitate sorting.

## 3. Results

### 3.1. Key Findings

#### Health Education during Pregnancy and Lactation on LRRFs in Gynecological and Obstetric Care

The analysis showed that gynecologists and midwives did not use a standardized procedure or set of concepts when educating pregnant or lactating women about LRRFs during pregnancy and lactation. The practice of health education about LRRFs is a complex phenomenon, influenced by factors associated with the current practice of health education about LRRFs, with respect to occurrences that lead to the development of the phenomenon (so-called “causal conditions” in the context of Strauss and Corbin’s coding paradigm framework). In this context, we investigated (a) causal conditions, including a woman’s medical history and specific needs regarding LRRFs, her socioeconomic status (SES), and migration background; (b) context (the set of characteristics in which a phenomenon is embedded), including the healthcare setting, timing of initial consultation, and time taken for consultation; and (c) intervening conditions (general conditions that influence action strategies), including barriers such as unfavorable regulations, unclear assignment of responsibility, and lack of standardized guidelines. We identified (d) two different action strategies (an action that is performed or not to overcome a phenomenon) applied by HCPs to educate women on LRRFs in gynecological and obstetric care, including a demand-driven healthcare approach and the differentiation of health education during pregnancy and lactation. As far as the (e) consequences (real or hypothetical) of the phenomenon go, we identified the necessity of support from other health organizations and pre-conception education of women on LRRFs.

To capture the phenomenon of health education during pregnancy and lactation on LRRFs in gynecological and obstetric care, each component of the coding paradigm (see Figure 2) is described in detail in the following sections.

If statements apply to both gynecologists and midwives, they are summarized as HCPs. The quotations were translated from German into English.

A summary of results is shown in Appendix A.

### 3.2. Causal Conditions

The results showed that the delivery of health education about LRRFs in gynecological and obstetric care is influenced by different causal conditions, including a woman’s medical history and specific needs regarding LRRFs, her socioeconomic status (SES), and her migration background.

#### 3.2.1. Medical History and Specific Needs

Some gynecologists and midwives mentioned that health education is given (or not) according to a woman’s medical history. A midwife mentioned as an example:

“If I have a woman in treatment and I ask during anamnesis for her experience with alcohol, drugs, and smoking and she denies everything and I have the impression that she is telling me the truth, then I don’t start to educate her that smoking is bad.” (M5)

Another example from a gynecologist supporting this concept follows: 

“So, we offer a consultation for at-risk patients who come specifically with a question or a pre-history. Therefore, we are very oriented to the specific problem at hand.” (G7)

Most HCPs reported that they listen to women’s self-expressed needs and questions. Health education is then tailored to the woman’s individual desires, questions, and specific needs. One midwife reported that she offers extra appointments to discuss issues in detail if the woman has a specific need:

“So, I always offer an additional appointment to a woman so that if she has any complaints she can come back and we can work on the issue then.” (M13)

Conversely, HCPs typically reported that health education is not given if there is no specific need, problem, or question; in other words, health education is given only if there is a need. The following statement is exemplary:

“If a pregnant woman has no problems, is content and happy, then I do not need to educate her much.” (G9)

#### 3.2.2. Socioeconomic Status

Almost all HCPs reported that the perceived necessity of health education about LRRFs in gynecological and obstetric care among women during pregnancy and lactation depended on the woman’s SES, understood as the level of education, occupation, income, and sometimes living area.

Most HCPs reported that they deliver health education when women actively communicate LRRFs or when they obviously expose the unborn child to LRRFs. HCPs assumed that health education is more intensive among less-educated women and, in this population, health education content needs reinforcement. Moreover, most HCPs reported that women from socially deprived areas have a much higher load of LRRFs compared to well-off women. Well-educated women were reported to have healthier lifestyles. One gynecologist reported that well-educated women are more likely to stop adverse health-related behaviors during pregnancy compared to less-educated women:

“There is the teacher who smoked occasionally and stops as soon as she is pregnant [whereas] the less-educated patients always come up with a standard argument, such as that the child might go into withdrawal.” (G3)

However, a midwife reported that less-educated women are more likely to disclose LRRFs, in contrast to women with a higher SES level:

“What I notice quite clearly, again and again, is: the more intelligent, the richer, the more distinguished the people, the more the alcohol is concealed. So, this very simple-minded woman tells me much sooner that she has eaten half a box of Mon Chéri [liquor-filled confectionery] than the well-off factory owner’s wife who likes to drink her glass of wine in the evening.” (M7)

The woman’s SES was identified to be a relevant factor in deciding whether and to which intensity health education about LRRFs during pregnancy would be delivered by the HCP.

#### 3.2.3. Migration Background

Besides SES, migration background was perceived to be a relevant factor in the practice of health education about LRRFs in gynecological and obstetric care, according to the interviewees. Most HCPs reported that women with a migration background have a higher risk of LRRFs during pregnancy. Several HCPs reported that Turkish women have a higher risk of unhealthy nutrition and being overweight or obese. HCPs identified higher risks of smoking and drinking among women from specific cultures. For example, one midwife stated:

“So, culturally, it depends very much on where the women come from, also in terms of smoking. African women smoke very little, maybe they smoke pot, but they don’t smoke cigarettes, but they drink more alcohol. […] if I have a Russian family for example, it’s completely normal that vodka is already poured out in the morning. It depends a bit on where the women come from, whether they drink or not, but during pregnancy they usually don’t, unless they have an addiction problem.” (M7)

Most HCPs reported that women with migration backgrounds specifically need to be educated on LRRFs, but they often adhere to their unhealthy habits although they know that it is unfavorable for the offspring.

Women who do not speak or understand German well cannot receive verbally communicated health education.

### 3.3. Context

Health education about LRRFs in gynecological and obstetric care was found to be context-dependent. Specifically, the healthcare setting (inpatient or outpatient), the date of the initial meeting, and individual schedules varied.

The delivery of health education about LRRFs during pregnancy and lactation varied between inpatient and outpatient healthcare settings. In inpatient healthcare settings, some gynecologists reported that they do not educate women on LRRFs during their visits, as stated here:

“We don’t do that. So, we don’t do any education there.” (G7)

However, in specific inpatient healthcare settings, health education is given systematically and in an organized institutional context where the partner is included in so-called “information evenings for parents,” as one gynecologist explained:

“I then hold information evenings and 100 people come. So usually, the women come with their husbands. And the danger of alcohol consumption during pregnancy takes up a large part. When they go home, they all understand that. I then also say that everything else is to be reiterated.” (G1)

Midwives who worked in a delivery room only reported that this is not the proper setting to educate women on LRRFs:

“Well, I do relatively little [laughs] education in pregnancy. I work in the delivery room, where women usually give birth.” (M8)

The data analysis showed that, in inpatient healthcare settings, individual education about LRRFs is given a lower priority compared to in outpatient healthcare settings. In outpatient healthcare settings, all HCPs stated that health education about LRRFs is given. The extent of the educational consultation varied greatly among outpatient HCPs in terms of content and time taken.

#### 3.3.1. Date of Initial Consultation

Most HCPs stated that education about LRRFs takes place at the initial consultation between the HCP and the pregnant women. However, the date of the initial meeting was reported to be subject to strong variation depending on the progress of the pregnancy. This also depended on the healthcare setting (inpatient versus outpatient). In inpatient care settings, with the exception of birth centers (where the first consultation is often before the 10th week of pregnancy), the date of the initial meeting was significantly later, between the 20th and 40th week of pregnancy, as compared to outpatient healthcare settings, where the first consultation takes place from the 5th week of pregnancy onwards.

#### 3.3.2. Time taken

The time taken for education about LRRFs varied among HCPs. Some HCPs reported that they advise women on LRRFs for a maximum of 5 min; others said that it takes about 1 h. However, in general, midwives reported spending substantially longer than gynecologists. The time taken also varied depending on the topic: HCPs dedicated more time to education about LRRFs during pregnancy compared to LRRFs during lactation. However, there is no fixed time needed to educate women on LRRFs, as a gynecologist stated:

“I always take time for the pregnant women [...] sometimes it’s longer, sometimes it’s a little less time.” (G2)

The intensity of education about LRRFs was reported to also depend on the HCPs’ individual schedules:

“So, I can’t do much on the subject myself, sometimes because of my own time pressure.” (M2)

The time taken was also said to be defined by the time billable to the health insurance company.

### 3.4. Intervening Conditions

We identified several intervening conditions (general conditions that influence action strategies) in the practice of health education about LRRFs, especially barriers such as lack of standardized guidelines, unfavorable regulations, and unclear assignment of responsibility.

#### 3.4.1. Lack of Standardized Guidelines

A central barrier to education about LRRFs in gynecological and obstetric care is the lack of standardized guidelines or any official documents that can be used. Some HCPs mentioned that they had designed their own checklists for women’s medical histories. Most HCPs mentioned that they did not know of any guidelines:

“I actually have to pass. […] I don’t know if there are really any guidelines.” (M12)

Some HCPs explained that the maternity logbook serves as their guide. However, this book has no detailed content; it provides more of a basic framework of what to inform patients about. Some HCPs mentioned the maternity guidelines; however, they similarly reported that the existing maternity guidelines lack detailed content:

“The doctor and midwife have to educate […], but to the questions of “How to do this?”, “How to address it?”, and “How to get information?” […] there are no in-depth instructions.” (M11)

The data analysis showed that there are no standardized procedures and documents applied in the education about LRRFs in gynecological and obstetric care. Some HCPs had worked out their own content, but most of the surveyed HCPs reported a need for detailed content to adequately provide education about LRRFs in gynecological and obstetric care.

#### 3.4.2. Unfavorable Regulations

HCPs reported that their time available to educate women on LRRFs during pregnancy and lactation is limited due to the billing options of the health insurance companies. The relevance of the topic of LRRFs during pregnancy and lactation is rated highly; however, the regulations regarding settlements with health insurance companies are unfavorable. Some midwives also reported that they deliver education about LRRFs in their own time, as the information is important but time for education is not fully covered by the health insurance company. Because of these unfavorable regulations, some midwives reported that they need to prioritize the topics that come up during individual consultations. Information on LRRFs during lactation might be a relevant topic during puerperium; however, the unfavorable regulations hinder provision of adequate education about LRRFs during lactation:

“Actually, you wouldn’t have any time for it during the puerperium. Because the puerperium is very strictly regulated in terms of payment, technically covering only 20–30 min […] what is more, I don’t get paid; that’s my personal service.” (M9)

#### 3.4.3. Unclear Assignment of Responsibility

Most of the HCPs reported that there is a particular authority figure or institution that should deliver education about LRRFs during pregnancy and lactation in the case of an obvious problem or need. The following quote shows that, in the case of specific problems, another institution might be responsible:

“There is a specialist […], another counseling center […]. Because that would go beyond my scope.” (M4)

None of the HCPs described themselves as the main person responsible for educating women on LRRFs, although they still do it—at least to some extent.

We identified an unclear assignment of responsibilities regarding education about LRRFs during pregnancy and lactation. One midwife explicitly stated that it is unknown whether education about LRRFs is performed by gynecologists.

“Basically, this is one of the important topics that should take place in the gynecological practice as well as in midwifery care, … but unfortunately I don’t know if it is included in every gynecological practice.” (M12)

### 3.5. Action Strategies

We identified two different action strategies among HCPs in the implementation of education about LRRFs: (1) a demand-driven healthcare approach and (2) differentiation between health education during pregnancy and lactation.

#### 3.5.1. Demand-Driven Healthcare Approach

The HCPs applied a demand-driven healthcare approach regarding education about LRRFs. Most HCPs reported that they review the maternity logbook to see whether women have already received information in accordance with this resource. If there are any gaps or open questions, these are then addressed individually.

Accordingly, the healthcare-related needs and circumstances of the women are determined during consultation. Specifically, a woman’s medical history guides the direction and focus of the session. Several HCPs reported a major need for HE regarding overweight, obesity, and diabetes, as it is recognized as an increasing public health problem in the pregnant women that appear in the HCPs’ care setting. Most HCPs reported that physical inactivity is very often observed, as shown in this statement: “There are many women who, for example, take no exercise at all.” (G2)

However, there was no consistency, as some HCPs reported that physical activity is not routinely a subject of conversation. Specifically, women with a medical history of (gestational) diabetes were reported to receive information on physical activity and nutrition. One gynecologist reported asking women about their physical activity and, in the case of high-risk activities such as skydiving or kickboxing, prohibited these activities.

Women with gestational diabetes are referred to diabetologists and nutritionists. Several HCPs reported that women with a special diet, for example vegan or vegetarian, also receive demand-oriented information regarding possible micronutrient supplementation. HCPs reported many uncertainties among pregnant women about what they should eat or avoid during pregnancy, e.g., one midwife reported that one of her clients boiled lettuce. Only a few HCPs reported that they talk routinely about what should be eaten and what foods should be avoided during pregnancy.

Most HCPs reported that they rarely encounter women who obviously consume alcohol during pregnancy. A few HCPs reported that they often see women who consume alcohol during pregnancy with a blood alcohol concentration of 0.30%. These women are then referred to addiction counseling services to address their specific healthcare-related needs. However, one HCP reported that most women believe that the consumption of a little bit of alcohol does not cause any harm. The data showed again that there is no standardized procedure and that alcohol consumption is sometimes brought up—specifically if there is an obvious problem—but not routinely.

A midwife reported that “alcohol is actually an issue for every woman in the puerperium” (M9), as most women want to know “When can I have a glass of wine again?” (M9). Some midwives reported that they inform women about the adverse effects of alcohol during lactation, but most HCPs reported that the issue does not arise at all.

Similarly, smoking during pregnancy is specifically addressed if there is a history of smoking. HCPs reported that they can often smell whether a woman smokes and encourage her to reduce smoking instead of stop smoking altogether:

“I’d rather give her advice on how to reduce smoking so that she doesn’t have a guilty conscience, because this will bother her even more.” (M10)

Specifically, smoking women are encouraged by HCPs to breastfeed the child, as the benefits of breastfeeding outweigh the adverse effects of smoking.

Several HCPs reported that stress during pregnancy is the most important lifestyle factor during pregnancy and after childbirth, and HCPs reported seeing many exhausted and stressed women. The occurrence of early labor is often associated with stress; one gynecologist outlined the procedure as follows:

“When we see someone in preterm labor, this is also explicitly addressed with the request either that we take this stress factor out of the equation by admitting the women as inpatients, or I sometimes tell the women to take it down a notch.” (G8)

HCPs specifically reported that women with more senior job positions have higher levels of stress. To reduce higher levels of stress, both midwives and gynecologists reported offering stressed women specific interventions such as qigong, respiratory therapy, etc. Some midwifes reported prescribing relaxation exercises, such as meditation, to stressed women, and giving specific advice on how to reduce stress. One midwife reported discussing stress resources during anamnesis to pinpoint specific stress factors. Depending on the result, the issue can then be addressed in the next appointment. This implies that the interviewed midwife developed an approach to screen women regarding stress factors and address these as necessary.

When HCPs perceived signs of stress, they openly addressed the issue and either acted or made recommendations for action to reduce levels of stress.

#### 3.5.2. Differentiation between Health Education during Pregnancy and Lactation

Across the study, there was an inconsistency of opinion on whether there are differences between health education during pregnancy and lactation. About half of the surveyed HCPs said there is no difference and the content is similar, as shown in the following statement:

“I would say they are similar in terms of addiction and such stuff.” (M3)

Accordingly, these HCPs do not provide women with different information specific to pregnancy and lactation.

However, the other half of HCPs surveyed reported that there is different advice given during pregnancy as compared to lactation. For example, nutritional advice is different in pregnancy compared to lactation.

Some midwives informed women that they should continue to avoid smoking and pay attention to their diet after giving birth. Some HCPs noted that lactation is so important that even smoking women should breastfeed their babies, because the positive effects of breastfeeding outweigh the negative effects of smoking.

“So, I tell them that they should rather smoke and breastfeed because breastfeeding is very important.” (G8)

As a result, the statements of the surveyed HCPs regarding differentiation between health education during pregnancy and lactation were inconsistent.

### 3.6. Consequences of Current Practice

#### 3.6.1. Pre-Conception Education of Women on LRRFs

HCPs indicated that contact with HCPs often occurs too late to educate women on LRRFs. Thus, they suggested educating women on LRRFs as early as possible. Both gynecologists and midwifes stated that the education of women on LRRFs during pregnancy and lactation should take place prior to conception, ideally in the early years of puberty during the transition into childbearing age. One midwife said that education about LRRFs should be part of periconceptional primary care.

There were different suggestions on the setting—for example, in gynecological care, in school, or even in preschool. This, however, implied that the surveyed HCPs would like to shift the responsibility for education on LRRFs to other (nonhealthcare) professionals.

#### 3.6.2. Support from Further Health Organizations

HCPs stated that support should be provided by other health organizations and that these organizations should educate women about LRRFs during pregnancy.

HCPs reported that the education about LRRFs needs to be designed to be more attractive for HCPs regarding their time and compensation. Moreover, financial support to deliver health education about LRRFs was mentioned by many HCPs. “If there were more money for billing […] then I could imagine that more education would be provided.” (M9)

One gynecologist suggested that the regulations should be changed so that midwives can accompany women at the beginning of a pregnancy, deliver education about LRRFs, and be adequately paid for this service. Adequate payment and a change to the payment regulations were supported by several HCPs.

HCPs reported that they give health education in their own time, and those regulations therefore need to be changed. Support from other health organizations seems to be an important factor in improving billing options for midwives and gynecologists. The relevance of educating women on LRRFs was rated highly, and HCPs stated that it should start as early as possible and be integrated into diverse settings. Health education about LRRFs during pregnancy and lactation requires more practitioners, improved cooperation between HCPs, and support from further health organizations.

## 4. Discussion

### 4.1. Key Findings

Most HCPs attested to the high relevance of education about LRRFs. However, in practice, education about LRRFs during pregnancy and lactation does not follow a standardized procedure and is often coincidental and sparse. In many inpatient healthcare settings, HCPs report that health education is a lower priority compared to in outpatient healthcare settings, especially because most women attend inpatient healthcare facilities in late pregnancy, whereas education about LRRFs during pregnancy should take place at the beginning of pregnancy. Inpatient care in Germany is usually utilized by pregnant women for whom certain risk factors are known or if complications occur during pregnancy. This might explain why, in these particular cases, HCPs perceive other issues as more pressing and important. However, most inpatient healthcare centers do host an information evening for parents-to-be where LRRFs during pregnancy and lactation can be addressed, and some reported that they do address these issues already.

HCPs currently apply a demand-driven healthcare approach with an individual thematic focus, guided by the woman’s medical history, needs, and circumstances. At the same time, some HCPs assign general stereotypes to certain patient groups, assuming specific lifestyle-related behaviors and thus specific needs for health education among low-SES women and migrants.

HCPs mentioned a lack of time and guidance, and unfavorable regulations and financing, as factors that prevented them from routinely educating women about LRRFs. In addition, most of the surveyed HCPs did not feel they were the main person responsible for educating women about LRRFs during pregnancy and lactation, and stated that this role should be played by other institutions such as schools. They spoke of a need for support from other health organizations to ensure early education and revised policy regulations. Health education about LRRFs during pregnancy and lactation needs to be a billable service to improve HCPs’ willingness to fully cover the topic during primary care, and a clear designation of persons responsible for ensuring that pre-conception health education about LRRFs during pregnancy and lactation is delivered. Moreover, there is a need for concrete guidelines that ensure evidence-based and substantiated health education about LRRFs during pregnancy and lactation.

While health education about LRRFs is deemed important and is often delivered by HCPs to a certain extent, the responsibility for this task remains largely unclear and needs to be clarified in the future.

### 4.2. Discussion of the Key Findings

In this study, HCPs stated that they do not routinely deliver education about LRRFs during pregnancy and lactation in gynecological and obstetric care due to a lack of time, a lack of guidelines, unfavorable regulations, and undefined responsibilities. Other studies have reported similar results, including a lack of time specifically in preventive reproductive healthcare [41], and for prevention and education in primary care in general [42,43,44,45]. As a reason for limited health education, HCPs reported a lack of guidelines. This finding agrees with other studies on health education in primary care [44] and in cancer care [46]. The HCPs reported that there are no standardized concepts, detailed guidelines, or checklists to be applied in health education about the overall topic of LRRFs during pregnancy and lactation in Germany in routine care. However, there have been preliminary efforts to address specific LRRF subjects during pregnancy, such as overweight, obesity, and malnutrition, in prenatal counseling sessions: In the context of a study in five Bavarian regions in Germany, the GeliS study (“Gesund leben in der Schwangerschaft”/Healthy living in pregnancy) implemented and evaluated lifestyle counseling with a focus on nutrition, physical activity, and weight monitoring during pregnancy and postpartum through midwives, gynecologists, and medical personnel [47]. The study had positive effects on dietary behaviors during pregnancy [48] and on offspring health outcomes [49]. The World Health Organization has developed recommendations for antenatal care in the English language, including some aspects related to LRRFs, such as nutrition [50]. Moreover, the Germany-wide Healthy Start—Young Family Network established national consensus recommendations on lifestyle before and during pregnancy as well as in the first years of life [51,52,53]. These recommendations contain detailed counseling content on lifestyle and are supported by the professional associations of gynecologists, pediatricians, and midwives. Hence, in theory, a common basis for counseling on LRRFs in the prenatal period exists, even though it is obviously not recognized and implemented in practice. The interviewed HCPs did not mention the existence of any recommendations for lifestyle counseling during pregnancy, and presumably they do not know about their existence. Even in our focus group discussion, the experts emphasized the urgent need of clear guidelines due to a lack of valid information. Overall recommendations exist, but guidelines on LRRFs during pregnancy, to be implemented in routine gynecological and obstetric care, ideally in multiple languages, are not available at the time of writing. A lack of obligation and missing billing possibilities could be potential reasons that existing recommendations are not implemented in daily practice. The creation of a clinical guidance with pertinent recommendations for HCPs that guarantees every pregnant woman lifestyle counseling during pregnancy might be a good start for the improvement of gynecological and obstetric care during pregnancy.

Due to the mentioned lack of clear guidelines, checklists, and an overall vision, as well as the mentioned barriers to implementation, most HCPs currently apply a demand-driven healthcare approach with an individual thematic focus, guided by the woman’s medical history, needs, and circumstances. The demand-driven healthcare approach focuses on the individual needs of the patient as considered by the HCP. Thus, the information provided to the pregnant woman is based on the HCPs’ perception of the woman’s knowledge. As we identified large knowledge gaps among a mainly well-educated study population in a previous study [37], the implementation of comprehensive lifestyle counseling during pregnancy is needed for knowledge increase and behavior change in the targeted population. However, counseling may be spread over various sessions and providers, for instance because in-depth lifestyle counseling is necessary (i.e., in case of special diet such as vegan diet, or overweight, obesity or substance addictions, etc.). The involvement of other providers, such as dietitians, might contribute to a relief of HCPs and increase the provision of pregnant women’s healthcare. Filby et al. reported that midwifery care is subject to unfavorable regulations and conditions, including economic barriers such as low or absent wages, a lack of governmental financial commitment, and informal payments, as well as professional and social barriers [54]. These barriers hinder the delivery of high-quality midwifery care [54]. Similar to other studies’ findings [55,56], we found that undefined and unclear responsibilities are barriers to the delivery of health education. A systematic review found that this also applies to the collaboration between midwives and nurses in birth care, as the collaboration is hindered by distrust, undefined roles, and a lack of professionalism and consideration [57]. There is a need for clearly defined responsibilities and role clarity between HCPs in gynecological and obstetric care. Strategies need to be implemented to address and decrease barriers in the collaboration between HCPs [57], which might improve gynecological and obstetric care.

Furthermore, some HCPs assign general stereotypes to certain patient groups, assuming specific lifestyle-related behaviors and thus specific needs for health education among low-SES women, migrants, and women with a concerning medical history and/or specific needs. There are, in fact, studies that confirm these assumptions by reporting higher rates of smoking, physical inactivity, and malnutrition among low-SES women and migrants [58,59,60]. However, many studies report that the prevalence rate of alcohol intake during pregnancy is higher among women with a higher SES [61,62,63,64]. A previous study conducted by the authors showed that a large knowledge gap on LRRFs during pregnancy exists across all SES levels [37]. Therefore, education about LRRFs during pregnancy and lactation should address women across all SES levels, and HCPs should be trained in educating higher-SES women as well.

A systematic review reported that the pre-conceptional period is a sensitive and important phase during fertility counseling [65]. Small modifications in the individual’s health behaviors can positively impact fertility and pregnancy outcomes [66]. However, most women do not receive adequate pre-conception advice during gynecological and fertility care [66]. It is recommended to develop and implement guidelines and checklists for HCPs to educate women on LRRFs prior to conception [65]. A systematic review reported that pre-conception education and counseling resulted in better knowledge, greater self-efficacy and a health-related locus of control and health risk behavior [67]. A study across six European countries found that all countries had pre-conception recommendations for chronically ill women, such as those with diabetes. However, recommendations for healthy women and men were incomplete and inconsistent across countries. Inconsistencies were found specifically regarding nutritional advice, alcohol consumption, and nutrient supplements [68]. However, pre-conception health education was found to be less effective in multiparous and low-SES women; thus, a tailored interventional approach could be more effective [69].

We found a higher priority placed on health education during pregnancy and lactation in outpatient care. In inpatient care, HCPs reported that the focus is on delivery or pregnancy complications, e.g., dealing with premature labor. Moreover, HCPs suggested the inclusion of other health organizations and community-based institutions for education about LRRFs (including prior to conception). There have been great efforts to include other health organizations and implement community-based interventions during health education about LRRFs during pregnancy and lactation, specifically in lower-middle-income countries [70]. However, most HCPs did not feel themselves to be the main person responsible for educating women on LRRFs during pregnancy and lactation. Still, preventive education during pregnancy is part of primary care, and a German study showed that nutritional counseling by gynecologists, midwives, and medical staff is effective [48,49]. Preventive oral health interventions are implemented across the globe [71], but there are no standardized guidelines for health education in primary care and the implemented interventions are inconsistent in content [72]. A study in Maharashtra, India, suggested that intensive prenatal oral health education improves women’s knowledge, attitudes, and behaviors [22]. As health education is an effective tool to influence and modify health behavior, a multisetting and multicomponent approach might be helpful to increase efficacy.

### 4.3. Potential Strengths and Limitations

To the best of our knowledge, this is the first study to assess HCPs’ perspectives on health education about LRRFs in gynecological and obstetric care in Germany. This study examined barriers, beneficial factors, and solutions that may support future innovative healthcare approaches, and policy and intervention programs for the implementation of education about LRRFs during and after pregnancy into routine care and other relevant health education settings. We obtained responses from a broad spectrum of HCPs by surveying HCPs from different healthcare settings (inpatient and outpatient; gynecological and obstetric; birth centers, and specialized, routine, and basic care providers), different areas (rural, urban, socially deprived), different professions (gynecologists and midwives), and varied levels of experiences. Most of the surveyed HCPs rated the topic as relevant, recognized it as neglected, and were enthusiastic about contributing to future solutions in their own and other health education settings. Purposive sampling was successful as we were able to recruit HCPs from all predefined categories (profession, care setting, type of provider, and area).

Our study has some potential limitations that need to be discussed. The study’s setting was limited to the federal state of Baden-Wuerttemberg, Germany. The practice of health education in general might vary among countries as the regulations and financial context surrounding health education might vary. Therefore, our results may not apply to other states or countries.

We included midwives in inpatient care. Some of them are in contact with women solely in the delivery room. This was not clear from the beginning of the interview. Although midwives in the delivery room consider the topic of LRRFs as relevant, the relevance of LRRFs at their work in the delivery room is of course of lower priority due to the higher priority of the delivery itself.

We interviewed twice as many HCPs in urban areas as in rural areas. Socially deprived areas were underrepresented, with two interviews only. We interviewed more midwives (13) than gynecologists (9). A comparison of the responses between HCPs in urban, rural, and socially deprived areas was not possible. Potential selection bias might have occurred, as HCPs who are interested in the topic might have been more likely to participate in the study. Thus, the results on the delivery of health education in gynecological and obstetric care could be biased, as HCPs who consider the topic to be relevant might be more likely to provide their patients with health education about LRRFs.

## 5. Conclusions

LRRFs during pregnancy and lactation are an increasing public health problem, posing a significant economic burden in lower-middle-income countries, upper-middle-income countries, and high-income countries. Reducing LRRFs during pregnancy and lactation requires adapted policy regulations, defined responsibilities for education about LRRFs during pregnancy and lactation, new guidelines and education tools to be integrated in routine care, and the definition of other potential settings for health education. Many HCPs, specifically in outpatient healthcare settings, are keen to provide information and educate women about LRRFs during pregnancy and lactation through the application of a demand-driven healthcare approach. The responsibility of HCPs to educate women on LRRFs needs to be strengthened in order to successfully implement lifestyle counseling during pregnancy and lactation into gynecological and obstetric care. This study highlights the need to understand the complexity of factors associated with education about LRRFs in gynecological and obstetric care and provides recommendations to improve the quality of this care. Health education in gynecological and obstetric care, policy regulations, and responsibilities differs among countries; further research is thus required as to how best to provide this care across lower-middle-income countries, upper-middle-income countries, and high-income countries.

## Figures and Tables

**Figure 1 ijerph-19-11674-f001:**
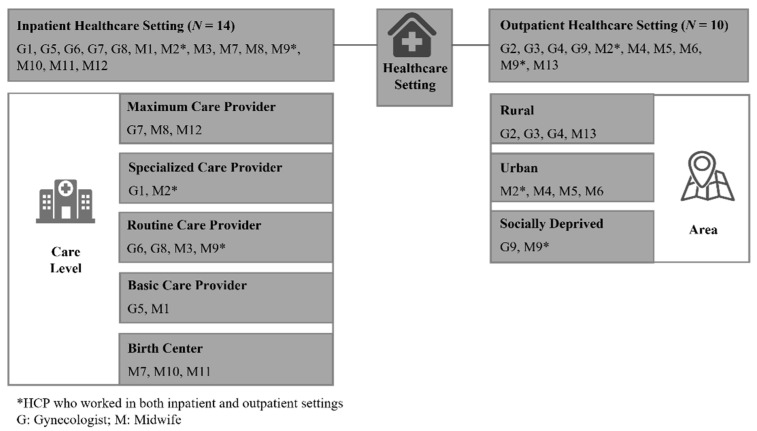
Study sample.

**Figure 2 ijerph-19-11674-f002:**
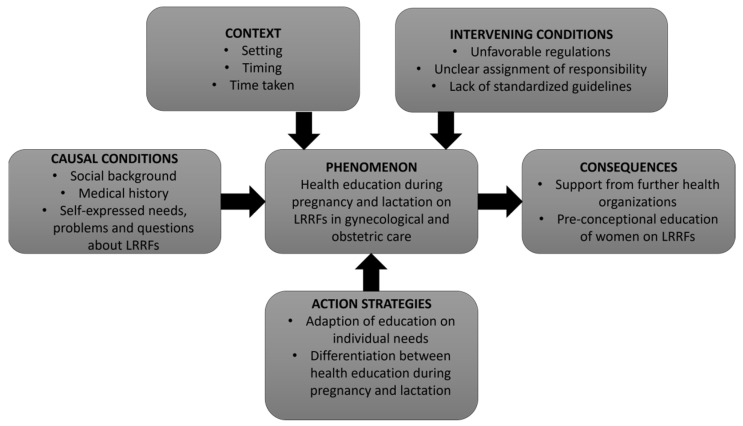
Coding paradigm for health education about lifestyle-related risk factors during pregnancy and lactation in gynecological and obstetric care from the perspective of HCPs.

## Data Availability

The data that support the finding of this study are available from the corresponding author (M.B.) upon reasonable request. The data are not publicly available due to ongoing studies.

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
