# Peer review of "Health Education about Lifestyle-Related Risk Factors in Gynecological and Obstetric Care: A Qualitative Study of Healthcare Providers’ Views in Germany"

_ijerph, 2022, doi:10.3390/ijerph191811674_

Round 1

Reviewer 1 Report (Previous Reviewer 1)

Dear authors, 

the resubmitted manuscript seems to be substantially improved after the previous round of revisions and is now suitable for publication. 

Kind regards 

Author Response

Dear reviewer,

we wish to thank you for the positive evaluation of our manuscript.

Kind regards,

Manuela Bombana and co-authors

Reviewer 2 Report (Previous Reviewer 2)

An overall statistical figure summarized each content and aspect would facilitate the readers regarding the soundness of each content.

Author Response

We thank the reviewer for the positive evaluation of our manuscript. We followed the reviewer's suggestion and created a supplementary table that summarizes each content. 

This manuscript is a resubmission of an earlier submission. The following is a list of the peer review reports and author responses from that submission.

Round 1

Reviewer 1 Report

Dear Authors, 

thank you for reporting on this interesting study which covers a highly relevant topic. Please find enclosed my comments and suggestions. 

Author Response

Dear reviewer,

we wish to thank you for the carefulness of your review. We addressed all your comments in our point-by-point response below. We believe that the thorough revision of our manuscript on the basis of your and the other reviewers valuable comments improved the manuscript substantially.

Please find enclosed our point-by-point responses.

Reviewer 2 Report

The manuscript described the role of health education about Lifestyle-related risk factors during pregnancy and lactation in gynecological and obstetric care from healthcare providers’ point of view. The combined inductive and deductive approach applied in this study explored and emphasized the need for a central strategy for health education about Lifestyle-related risk factors during pregnancy and lactation in gynecological and obstetric care. The evidence is well organized to support the summary of the needing of central strategy for health education.

1.       Some of the words lack explanation, e.g., HE (Section 2.3, Page 4)

2.       There are some redundancy descriptions, e.g., Section 2.1 versus Section 2.2 for the population descriptions.

3.       A parallel solution workflow figure compared with Figure 2, for discussion section (Section 4.1 and Section 4.3, Page 11-13), would facilitate the potential readers.

Author Response

(The authors gave the same response as above.)

Round 2

Reviewer 1 Report

Dear authors, 

please find enclosed my comments on the revised version of your manuscript. 

Best regards 
